# In Vitro Antioxidant and Antitrypanosomal Activities of Extract and Fractions of *Terminalia catappa*

**DOI:** 10.3390/biology12070895

**Published:** 2023-06-22

**Authors:** Sandra Alves de Araújo, Aldilene da Silva Lima, Cláudia Quintino da Rocha, Henrique Previtalli-Silva, Daiana de Jesus Hardoim, Noemi Nosomi Taniwaki, Kátia da Silva Calabrese, Fernando Almeida-Souza, Ana Lucia Abreu-Silva

**Affiliations:** 1Rede Nordeste de Biotecnologia, Universidade Federal do Maranhão, São Luís 65080-805, MA, Brazil; aasandra09@gmail.com (S.A.d.A.); abreusilva.ana@gmail.com (A.L.A.-S.); 2Laboratório de Química dos Produtos Naturais, Universidade Federal do Maranhão, São Luís 65080-805, MA, Brazil; 3Laboratório de Protozoologia, Instituto Oswaldo Cruz, Fiocruz, Rio de Janeiro 21041-250, RJ, Brazil; henriqueprevitale@hotmail.com (H.P.-S.); hardoim@ioc.fiocruz.br (D.d.J.H.); 4Núcleo de Microscopia Eletrônica, Instituto Adolfo Lutz, São Paulo 01246-000, SP, Brazil; ntaniwak@hotmail.com; 5Pós-Graduação em Ciência Animal, Universidade Estadual do Maranhão, São Luís 65055-310, MA, Brazil

**Keywords:** chagas disease, *Trypanosoma cruzi*, parasite, ellagic acid, alternative treatment, combretaceae

## Abstract

**Simple Summary:**

Chagas disease is a serious infection caused by an intracellular parasite and transmitted primarily through an infected insect. Despite available treatments, the disease continues to cause countless deaths around the world due to the ineffectiveness of drugs. Medicinal plants have been used as an alternative and effective treatment against various diseases. In this work, we verified the antioxidant properties of the extract and fractions obtained from the plant species *Terminalia catappa* and its action against the parasite responsible for Chagas disease, *Trypanosoma cruzi*. Initially, we observed that the ethyl acetate and aqueous fraction demonstrated antioxidant activity. In addition, the ethyl acetate fraction showed the best inhibitory activity against all cellular forms of the parasite (epimastigotes, trypomastigotes and intracellular amastigotes), and did not present toxicity to host cells. We also observed that the ethyl acetate fraction induced several morphological changes to the parasite, such as cytoplasmic disruption, cell disorganization, morphological variation and loss of integrity. In this sense, we conclude that the ethyl acetate fraction obtained from *T. catappa* leaves can be an effective alternative in the treatment and control of Chagas disease.

**Abstract:**

Chagas disease is a severe infectious and parasitic disease caused by the protozoan *Trypanosoma cruzi* and considered a public health problem. Chemotherapeutics are still the main means of control and treatment of the disease, however with some limitations. As an alternative treatment, plants have been pointed out due to their proven pharmacological properties. Many studies carried out with *Terminalia catappa* have shown several biological activities, but its effect against *T. cruzi* is still unknown. The objective of this work is to evaluate the therapeutic potential of extracts and fractions obtained from *T. catappa* on the parasite *T. cruzi*, in addition to analyzing its antioxidant activity. *T. catappa* ethyl acetate fraction were produced and submitted the chemical characterization by Liquid Chromatography Coupled to Mass Spectrometry (LC-MS). From all *T. catappa* extracts and fractions evaluated, the ethyl acetate and the aqueous fraction displayed the best antioxidant activity by the 2,2-diphenyl-1-picryl-hydrazyl (DPPH) radical scavenging method (IC_50_ of 7.77 ± 1.61 and 5.26 ± 1.26 µg/mL respectively), and by ferric ion reducing (FRAP) method (687.61 ± 0.26 and 1009.32 ± 0.13 µM of Trolox equivalent/mg extract, respectively). The ethyl acetate fraction showed remarkable *T. cruzi* inhibitory activity with IC_50_ of 8.86 ± 1.13, 24.91 ± 1.15 and 85.01 ± 1.21 µg/mL against epimastigotes, trypomastigotes and intracellular amastigotes, respectively, and showed no cytotoxicity for Vero cells (CC_50_ > 1000 µg/mL). The treatment of epimastigotes with the ethyl acetate fraction led to drastic ultrastructural changes such as the loss of cytoplasm organelles, cell disorganization, nucleus damage and the loss of integrity of the parasite. This effect could be due to secondary compounds present in this extract, such as luteolin, kaempferol, quercetin, ellagic acid and derivatives. The ethyl acetate fraction obtained from *T. catappa* leaves can be an effective alternative in the treatment and control of Chagas disease, and material for further investigations.

## 1. Introduction

Chagas disease is a serious infectious disease caused by the intracellular protozoan *Trypanosoma cruzi*. It is a neglected tropical disease, being one of the biggest public health problems in many countries, affecting about 6 to 7 million people worldwide [1,2,3]. In Brazil, it is estimated that at least 1 million people are infected with *T. cruzi* [4].

Currently, benznidazole and nifurtimox are the only drugs used to treat the Chagas disease [5,6]. However, these drugs are effective only in the acute phase of the disease, since that no improvement in the clinical manifestations is observed during the chronic phase [5]. In addition, the high incidence of toxicity and adverse effects compromise the continuity of drug use, most often leading to treatment abandonment [7]. Therefore, we emphasize the importance of developing new, effective treatment strategies in all stages of Chagas disease and minimizing the side effects of commercially available drugs.

Secondary metabolites obtained from plants have been evaluated due to their numerous medicinal and pharmaceutical properties, despite that few species have been scientifically studied to safely assess their qualities and efficacy. *Terminalia catappa* is a plant species belonging to the Combretaceae family, commonly known as “amendoeira” or “castanhola” [8,9]. In Brazil, it is found mainly in the Northeast region of the country [10]. Its leaves have biologically active compounds with some activities described in the literature, including anti-inflammatory and immunomodulatory [11], antidiabetic [12], antibacterial [13] and anticancer [14,15].

Although the numerous properties of *T. catappa* have been reported, there are no studies indicating its potential for the development of new products to control *T. cruzi*. Thus, this research aimed to study the action of extracts and fractions of *T. catappa* against extracellular and intracellular forms of *T. cruzi*, as well as to analyze their antioxidant properties and verify the direct effect on the parasite through ultrastructural analysis.

## 2. Materials and Methods

### 2.1. Plant Material

Samples of *T. catappa* leaves were collected in November 2021 at Fazenda Escola, State University of Maranhão in São Luís, Maranhão State, Brazil (Location: 2°35′08.7″ S; 44°12′30.7″ W). The plant was identified at Rosa Mochel Herbarium of State University of Maranhão (voucher specimen No. 5991). The Brazilian Genetic Heritage Management Council approved all procedures (Proc. No. AFC60DB).

### 2.2. Obtaining Extract and Fractions

The plant material was dried at room temperature (28 °C) for 10 days and then ground in a mill. The extraction was carried out by means of exhaustive percolation with ethanolic alcohol (850 g; 70% *v*/*v*) for 20 days. After extraction, the solvent was filtered and evaporated in a rotary evaporator with reduced pressure (40 °C) and subsequently lyophilized (−90 °C), obtaining the *T. catappa* hydroethanolic extract. For fractionation, the hydroethanolic extract (10 g) was diluted in methanol: water (400 mL; 8:2; *v*/*v*) solution in a liquid separating funnel, using solvents of different polarities, namely hexane, ethyl acetate and water. The solvents of each fraction were removed on a rotary evaporator (40 °C) and the yields were expressed as a percentage. All samples were stored at −20 °C until they were required for the biological assay. For cytotoxicity and antitrypanosomal activity assays, the extract and fractions were first diluted in DMSO and then diluted in a proper medium at a final DMSO concentration lower than 1%.

### 2.3. Chemical Characterization of T. catappa by HPLC-UV-ESI-IT/MS

Samples were analyzed in a Shimadzu Prominence liquid chromatography system with two Shimadzu LC-20ad (SIL-20a HT) (Shimadzu Corp., Quioto, Japan) automated injection pumps. Phenomenex Luna C18 column (250 × 4.6 mm − 5 μm) (Phenomex Inc., Torrance, CA, USA) was used to separate the components. The elution solvents used were A (0.1% HCOOH in acidified water) and B (0.1% HCOOH in acidified metanol/HPLC grade)—at a flow rate of 1.0 mL/min. Samples are eluted in a gradient system: 95% A/5% B-0 min-0% A/100% B-60 min and 100% B in 10 min with a runtime of 70 min. The liquid chromatogram was performed using electrospray ionization (ESI), multistage fragmentation (MSn) at an ion trap (IT) interface (Amazon X, Bruker, MA, USA) in the negative and positive mode under the following conditions: capillary voltage of 5 kV, capillary temperature of 325 °C, carrier gas flow (N_2_) of 12 L/min, and nitrogen nebulizer pressure at 10 psi. Full scan analysis was recorded in the M/Z range from 100–1500 *m*/*z*, with two or more events.

### 2.4. 2,2-Diphenyl-1-picrylhydrazyl (DPPH) Radical Scavenging Assay

The antioxidant activity of *T. catappa* was evaluated by scavenging the free radical 2,2-diphenyl-1-picryl-hydrazyl (DPPH) (Sigma-Aldrich, St. Louis, MO, USA). Initially, a solution of DPPH in methanol (0.3 mM) was prepared. In 96-well plates containing 100 μL of hydroalcoholic extract and fractions at different concentrations (500-1.95 μg/mL) diluted in methanol, 40 μL of DPPH solution was added. The plate was kept in the dark for 30 min and then read in a spectrophotometer at 492 nm. Wells with methanol and DPPH were used as negative controls. The standard curve (50-0.39 μM) obtained by (±)-6-Hydroxy-2,5,7,8-tetramethylchromane-2-carboxylic acid (Trolox) (Sigma-Aldrich, St. Louis, MO, USA) was used as a positive control. The assay was performed in triplicate. The percentage of DPPH radical scavenging of each concentration was used to calculate the required inhibitory concentration needed to eliminate 50% (IC_50_) of the DPPH for all samples.

### 2.5. Ferric Ion Reducing Antioxidant Power (FRAP) Assay

The ferric reducing antioxidant power (FRAP) reagent was prepared by mixing 25 mL 0.3 M acetate buffer (pH 3.6); 2.5 mL 10 mM 2,4,6-Tris(2-pyridyl)-s-triazine (TPTZ) (Sigma-Aldrich, St. Louis, MO, USA) in 40 mM hydrochloric acid (HCl) and 2.5 mL 20 mM iron chloride (FeCl_3_). The FRAP reagent was left at 37 °C for 30 min. In 96-well plates containing 10 μL of the hydroalcoholic extract and the fractions (at 500 µg/mL), 300 μL of FRAP reagent was added and 30 μL of deionized water. Then, the samples were incubated for 30 min at 37 °C. Absorbance of the samples was measured at 595 nm. A sample blank was prepared with FRAP reagent, methanol and deionized water. The assay was performed in triplicate. The absorbance sample was compared to the Trolox standard curve (1000-3.9 μM) and the results were expressed in μM Trolox equivalent (TE) per gram of extract (μM TE/mg extract).

### 2.6. Cell Culture and Parasites

The African green monkey kidney Vero cell line (ATCC CCL-81) was cultivated in Dulbecco’s modified eagle medium (DMEM) (Sigma-Aldrich, St. Louis, MO, USA), pH 7.2, at 37 °C and 5% CO_2_. *T. cruzi* (Y strain) epimastigotes forms were cultured at 28 °C in liver infusion tryptose (LIT) medium, pH 7.2. Trypomastigotes forms were obtained from infected Vero cells and cultured in DMEM, pH 7.2 at 37 °C and 5% CO_2_. All culture media were supplemented with 10% fetal bovine serum (FBS) (LGC Biotecnologia, São Paulo, Brazil), 100 U/mL penicillin (Sigma-Aldrich, St. Louis, MO, USA) and 100 µg/mL streptomycin (Sigma-Aldrich, St. Louis, MO, USA).

### 2.7. Cytotoxicity Assay

Initially, Vero cells were cultured in 96-well plates (5 × 10^5^ cells/mL) for 4 h and then, treated with *T. catappa*, ellagic acid (Sigma-Aldrich, St. Louis, MO, USA) or benznidazole at concentrations from 1000 to 31.2 µg/mL (100 µL/well) and incubated for 24 h in 37 °C and 5% CO_2_. Wells without cells and wells with cells and medium only were used as blank and control, respectively. The cell viability was determined by the colorimetric method using tetrazolium-dye 3-(4,5-dimethylthiazol-2-yl)-2,5-diphenyltetrazolium bromide (MTT) (Sigma-Aldrich, St. Louis, MO, USA), according to [16]. This experiment was carried out in triplicate and data was used to determine the 50% cell cytotoxicity (CC_50_).

### 2.8. Antitrypanosomal Activity Assay

In 96-well plates, *T. cruzi* trypomastigotes and epimastigotes forms (10^6^ parasites/mL) were incubated in different concentrations of *T. catappa* or ellagic acid for 24 h and 72 h, respectively. Concentration of *T. catappa* ranged from 500 to 1.95 µg/mL for the epimastigote forms assay and 500 to 7.8 µg/mL for the trypomastigote assay. Ellagic acid was evaluated at 500-15.6 µg/mL concentrations. After treatment, the viability of parasites was evaluated by counting the total number of parasites using a Neubauer chamber and light microscope. As a control, blanks were used with wells containing only parasites and wells without parasites, respectively. All samples were tested in triplicate. The results are expressed as parasite growth inhibitory concentration (IC_50_). For the intracellular amastigote experiment, Vero cells were cultured in 24-well plates (5 × 10 5 cells/mL) containing coverslips. Subsequently, the cells were infected with *T. cruzi* trypomastigotes (10:1 ratio, parasite/cell). After 6 h of infection, cells were washed to remove non-internalized parasites. In quadruplicate, the infected cells were treated with *T. catappa* (250-31.25 µg/mL) or benznidazole (100-6.25 µg/mL) for 24 h. The coverslips containing infected and treated cells with *T. catappa* were fixed in Bouin and stained with Giemsa (Sigma-Aldrich, St. Louis, MO, USA). After, coverslips were analyzed under light microscopy and the number of intracellular amastigotes in 200 cells was used to calculate the IC_50_. The ratio of Vero cells CC_50_/IC_50_ cells was obtained for the selectivity index (SI) calculation. The number of amastigotes per 200 cells, the percentage of infected cells and the number of amastigotes per infected cell were calculated as described elsewhere [17].

### 2.9. Transmission Electron Microscopy

*T. cruzi* epimastigotes forms were treated with IC_50_ of *T. catappa* ethyl acetate fraction for 72 h. The parasites were fixed overnight with 2.5% glutaraldehyde (Sigma-Aldrich, St. Louis, MO, USA) in a 0.1 M sodium cacodylate buffer (pH 7.2) at room temperature. After, the parasites were then washed three times with 0.1 M sodium cacodylate buffer, post-fixed in a solution containing 1% osmium tetroxide (Sigma-Aldrich, St. Louis, MO, USA), 0.8% potassium ferrocyanide (Sigma-Aldrich, St. Louis, MO, USA), and 5 mM calcium chloride for 60 min (Sigma-Aldrich, St. Louis, MO, USA). Then, the parasites were dehydrated in increasing concentrations of acetone (30–100%) for 10 min, embedded in epoxy resin EMbed-812 (Electron Microscopy Sciences, Hatfield, PA, USA) for 24 h at room temperature, and polymerized at 60 °C for 72 h. Untreated parasites were used as a comparison. Ultrathin sections (100 nm) were stained with uranyl acetate and lead citrate (Sigma-Aldrich, St. Louis, MO, USA) and observed under a Transmission Electron Microscope JEM-1011 (JEOL, Tokyo, Japan) operated at 80 kV.

### 2.10. Statistical Analysis

The IC_50_ and CC_50_ were obtained from a nonlinear regression fit curve of concentration log versus normalized response and the values were expressed as mean ± standard deviation performed with the software GraphPad Prism 7.0 (GraphPad Software, San Diego, CA, USA). The Mann–Whitney test was used to analyze the results and differences were considered significant when *p* < 0.05.

## 3. Results

### 3.1. Composition of Extract and Fractions of T. catappa

The hydroalcoholic extract from the lyophilized leaves of *T. catappa* yielded 18%, obtained from 850 g of vegetal mass. The fractionation of the extract (10 g), with hexane, ethyl acetate and water-methanol (7:3) provided the hexanic fraction (13.1%), ethyl acetate fraction (14.4%) and aqueous fraction (44.6%). The chemical characterization of *T. catappa* hydroalcoholic extract and ethyl acetate fraction performed by HPLC-UV-ESI-IT/MS is shown in Figure 1 and Table 1, respectively. Ellagic acid was identified as the major compound in the ethyl acetate fraction (Figure 1).

### 3.2. Antioxidant Activity of T. catappa

Extracts and fractions of *T. catappa* were evaluated by two different antioxidant methods: DPPH radical scavenging and FRAP ferric ion reducing. All samples were able to reduce DPPH radical and ferric ions (Fe^3+^) (Table 2). The aqueous and ethyl acetate fractions were the most potent in reducing the DPPH radical, compared to the Trolox standard, demonstrating its antioxidant capacity, as shown in Table 2. In the FRAP method, the hexane fraction reduced the Fe^3+^ radical.

### 3.3. Antitrypanosomal Activity, Cytotoxicity and Selectivity Index of T. catappa

The activity of anti-*T. cruzi* of *T. catappa* is shown in Table 3. All tested samples showed inhibitory effects against the different forms of the parasite. However, the IC_50_ values of the ethyl acetate fraction was lower compared to the other samples, exhibiting concentration-dependent activity against the three forms of *T. cruzi* evaluated (Figure 2). It was also observed that ellagic acid had an effect against epimastigotes forms, although its IC_50_ presented the highest value among the evaluated compounds, and no activity against trypomastigote forms. The cytotoxic activity and selectivity index (SI = CC_50_/IC_50_) of *T. catappa* are presented in Table 3. Hydroalcoholic extracts, ethyl acetate fractions and water fractions were not toxic to Vero cells at the concentrations evaluated, maintaining their cell viability even at the highest concentration. Similar results were observed for ellagic acid and benznidazole. The hexane fraction displayed the lowest CC_50_ value, resulting in a low SI against epimastigotes and trypomastigotes. The ethyl acetate fraction was the most selective against the three forms of the parasite (epimastigotes, trypomastigotes and intracellular amastigotes, respectively). The benznidazole also showed selectivity as expected.

The ethyl acetate fraction was selected for the experiments against intracellular amastigotes (Figure 3). Analysis of the parameters of infection indicated that the ethyl acetate fraction reduced the number of amastigotes per 200 cells at 62.5, 125 and 250 µg/mL (*p* < 0.01; Figure 3A), and decreased the percentage of infected cells (*p* < 0.01; Figure 3B) and the average number of amastigotes per infected cell (*p* < 0.01; Figure 3C) at 125 and 250 µg/mL. Benznidazole also showed a significant reduction in all infection parameters at all concentrations evaluated (*p* < 0.01; Figure 3D–F). The alterations in intracellular amastigote form of *T. cruzi* after treatment with ethyl acetate fraction are represented in photomicrography images of Figure 3G.

### 3.4. Effects of T. catappa on T. cruzi Morphology

The transmission electron microscope was performed to investigate the morphological changes induced by the *T. catappa* ethyl acetate fraction treatment and identify the intracellular targets in epimastigote forms of parasite (Figure 4). Untreated-epimastigotes presented elongated cell bodies with well-defined kinetoplasts and centrally located nuclei with evident nucleoli (Figure 4A). The treatment of epimastigotes with the ethyl acetate fraction led to drastic morphological changes (Figure 4B–D). The morphological alterations were kinetoplast swelling, disruption and loss of cytoplasmic organelles, detachment of the nuclear membrane (white arrow), with increasing nuclear damage; plasma membrane rupture (black arrow), with detachment of the lipid bilayer and loss of microtubule organization (Figure 4B–D).

## 4. Discussion

Due to the importance of controlling and treating Chagas disease, in this research we demonstrate for the first time, to the light of our knowledge, the action of secondary metabolites present in *T. catappa* against the *T. cruzi* parasite, which are able to cause ultrastructural alterations during the treatment with the ethyl acetate fraction.

Initially, we evaluated the antitrypanocidal activity of the hydroalcoholic extract and the hexane, ethyl acetate and aqueous fractions against the *T. cruzi* extracellular forms to define the most effective one. As a result, the ethyl acetate fraction exhibited a main activity against *T. cruzi* epimastigotes and trypomastigotes forms and did not exhibit cytotoxicity against Vero cells. Thus, the ethyl acetate fraction was used in further experiments.

Terminalia species have numerous phytochemical compounds responsible for different biological properties [8]. Quantitative analysis of the ethyl acetate fraction by HPLC-UV-ESI-IT/MS identified ellagic acid as the major compound. The identification of ellagic acid as the main compound has been reported in other *Terminalia* species, like *Te. avicennioides* [18], *Te. leiocarpa* [19] and *Te. molis* [20]. Ellagic acid is a phenolic compound naturally found in plants and fruits with several biological activities described as antioxidant [21], neuroprotective [22], anticancer [23,24], including its antitrypanosomal activity [20]. In this study, other phytochemicals identified can also be responsible for the anti-*T. cruzi* activity, namely luteolin, apigenin and quercetin, among others.

Plant-derived phenolic compounds are widely known for their antioxidant capacities, and these may be directly related to their biological activity [21,25]. An antioxidant is a compound that inhibits the production of free radicals. During *T. cruzi* infection, phagocytic cells induce an oxidative environment to control the parasites, producing reactive oxygen species (ROS), especially superoxide (O2^−^) and nitric oxide (NO), which together produce a strong oxidant and cytotoxic, peroxynitrite (ONOO^−^) [26,27]. According to Paiva et al. [28], the parasites are able to survive in the oxidative environment and the peroxynitrite radical can exert cytotoxic effects on host cells. Although the antioxidant effects of plant extracts against *T. cruzi* are still unclear, there is evidence that this action has the capacity to act on immune functions in organisms. Montenote et al. [29]. verified that *Morus nigra* extracts, rich in phenolic compounds, reduced parasitemia in the acute phase of Chagas disease, and that in the chronic phase they have activity on some antioxidant defenses, minimizing the tissue inflammatory process.

Conversely, the trypanocidal mechanism of benznidazole is attributed to its ability to generate oxidative damage to the parasite [30]. For Sánchez-Villamil et al. [31], a balance between ROS levels and the antioxidant response is fundamental in maintaining a safe environment for cells. The use of antioxidants as adjuvant compounds can reduce oxidative damage to host cells, becoming a potential in the treatment of Chagas disease [31,32]. Antioxidant properties of genus *Terminalia* have already been reported in the literature [22,33,34]. In this study, we verified the antioxidant potential of *T. catappa* extracts and fractions via scavenging of DPPH radicals and iron ions. In this manner, the antioxidant effect of *T. catappa* as an adjuvant therapy is interesting, since it can help to neutralize free radicals and protect cells against oxidative damage. However, it is still necessary to study the antioxidant mechanisms of *T. catappa* and correlate this with the bioactivity against *T. cruzi*.

Studies indicate that biological activities of plant extracts depend mainly on the plant part used, solvent and extraction method [35,36]. The use of solvents with different polarities has a great impact on the composition of the final extract [37]. Studies carried out by Rutin-Marie et al., observed that leaf extracts obtained with ethyl acetate (polar solvent) were more active against *Plasmodium falciparum*, indicating that ethyl acetate was the best solvent for the extraction of promising compounds [38].

The genus *Terminalia* has great potential in the development of effective drugs against protozoa. Ohashi et al. verified that *Te. ivorensis* hexane and ethyl acetate extracts showed antiprotozoal effects against *Trypanosoma*, *Leishmania* and *Plasmodium* species [39]. Additionally, Camara et al. [40] evaluated the in vitro and in vivo antiplasmodial activity of *Te. albida* extracts, observing the inhibition of parasitemia, absence of toxicity and greater animal survival, suggesting that these may be promising sources for the development of new drugs for controlling malaria.

In this research, *T. catappa* extract and fractions showed activity against the three forms of *T. cruzi*. Similarly, Griebler et al. demonstrated the effect of different extracts of *Lonchocarpus cultratus* against the three forms of the parasite, observing that the hexanic, dichloromethane and methanolic extracts showed trypanosomal activity with IC_50_ values of 15.5, 18.7 and 26.1 µg/mL after 24 h against trypomastigotes, respectively, while hexanic and dichloromethane extracts showed IC_50_ values of 26.7 and 4.8 µg/mL after 72 h against epimastigotes [41]. In addition, ellagic acid exhibited activity only against the *T. cruzi* epimastigotes forms, suggesting that existing structural differences in the epimastigotes and trypomastigotes forms may interfere with the bioactivity of the product.

Although assays against epimastigotes and trypomastigotes forms are important to trial compounds with antitrypanosomal activity, the main target is the intracellular form that patients present during the course of the disease [16]. The development of drugs that inhibit *T. cruzi* intracellular amastigotes is crucial in the treatment of the disease. In the literature, there are few reports of plant activity against *T. cruzi* intracellular forms. Therefore, it was of interest to test the efficacy of *T. catappa* against *T. cruzi* intracellular amastigotes.

Treatment of infected Vero cells with the ethyl acetate fraction of *T. catappa* at the highest concentrations (250-62.5 µg/mL) significantly affected the survival of intracellular amastigotes in the cells. Although the treatment of infected cells with the ethyl acetate fraction inhibited the number of parasites per 200 infected cells, the percentage of infected cells and the number of parasites per infected cell, the inhibitory concentration for 50% of parasites was not as low as expected when compared to the benznidazole (reference drug). Benznidazole showed trypanosomal activity as expected.

In murine macrophages infected with *T. cruzi*, Chaves et al. [42] evaluated the trypanosomal potential of extracts and fractions of *Manilkara rufula*, noting that the ethyl acetate fraction, methanolic fraction, hydroalcoholic fraction and hexane fraction induced the infectivity index reduction by 78.1, 53.9, 54.9 and 10.5%, respectively. The mechanism of action is still uncertain, although it is believed that the *M. rufula* fractions contain molecules with cytoprotective activity, especially the ethyl acetate fraction, which protected cells against H_2_O_2_-induced death [42].

The investigation of ultrastructural changes induced by *T. catappa* provides some evidence of its mode of action. Analysis by electron microscopy showed that the ethyl acetate fraction produced notable intracellular changes in *T. cruzi* epimastigotes. Based on our observations, the results point to the plasma membrane as possible target of the ethyl acetate fraction, indicating the loss of parasite integrity and, consequently, the loss of cytoplasmic organelles. Similar results were observed by De Melo et al. [5]; according to these authors, the *Lippia sidoides* and *Lippia origanoides* essential oils caused morphological changes in epimastigote forms compatible with the loss of viability and cell death of the parasite, indicating the mitochondria as the main target. In addition, ultrastructural damage of compounds isolated from plants on *T. cruzi* has also been reported. In experiments conducted by García-Huertas et al. and Londero et al., lignin (IC_50_ = 14.29 μg/mL) and costic acid (IC_50_ = 37.8 μM), respectively, caused intense ultrastructural damage in the cytoplasm, intracellular disorganization, presence of autophagic signals and formation of large vacuoles [2,43]. In bloodstream trypomastigotes, benznidazole (IC_50_ = 8.82 μM) caused vacuolation and disorganization of the cytoplasm [44]. These results suggest that intracellular structures are important targets in the development of drugs with antitrypanosomal activity. In the literature, no studies were found reporting structural damage in protozoa caused by extracts of *T. catappa*, especially in *T. cruzi*.

## 5. Conclusions

In conclusion, the present work demonstrates that the *T. catappa* extract and fractions have antitrypanosomal activity against *T. cruzi*. The absence of cytotoxicity associated with the excellent trypanocidal activity of the ethyl acetate fraction indicates a therapeutic potential of *T. catappa* for Chagas disease. In addition, the antioxidant capacity may be related to the efficiency of activity against the parasite. Although the mechanisms of action are still unknown, the death of the parasite may be directly related to structural damage, confirmed by electron microscopy. This study provides evidence that *T. catappa* leaves can be an effective alternative in the treatment and control of Chagas disease.

## 6. Patents

The patent BR 10 2023 006058 7 “Formulação farmacêutica à base de folhas de *Terminalia catappa* para tratamento das leishmanioses e doença de Chagas” was granted to Araújo, S. A.; Abreu-Silva, A. L.; Almeida-Souza, F.; Lima, A. S.; and Rocha, C. Q. by the National Institute of Industrial Property (Instituto Nacional da Propriedade Industrial—INPI) in 31 March 2023.

## Figures and Tables

**Figure 1 biology-12-00895-f001:**
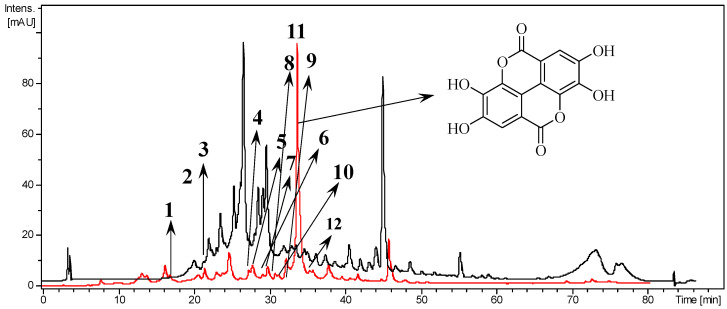
Chromatogram of *Terminalia catappa*. Hydroalcoholic extract (black) and ethyl acetate fraction (red). Chemical structure of ellagic acid (arrow).

**Figure 2 biology-12-00895-f002:**
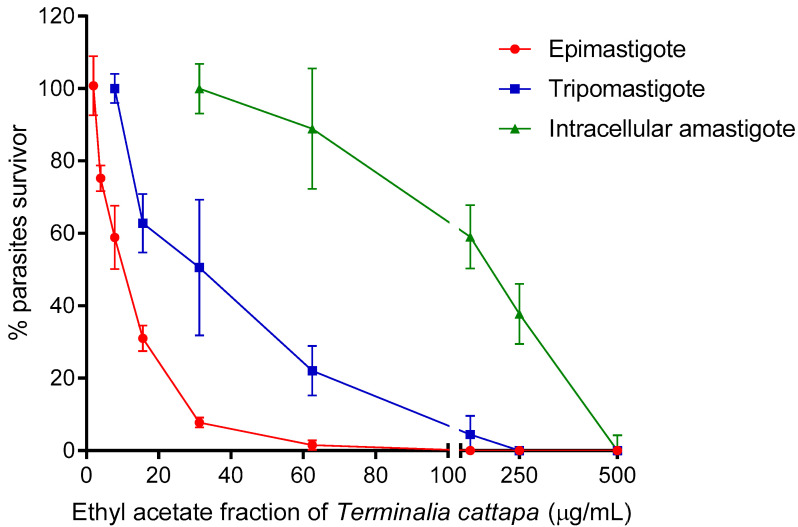
Concentration-response curve of ethyl acetate fraction of *Terminalia catappa* against epimastigote, trypomastigote and intracellular amastigote forms of *Trypanosoma cruzi*.

**Figure 3 biology-12-00895-f003:**
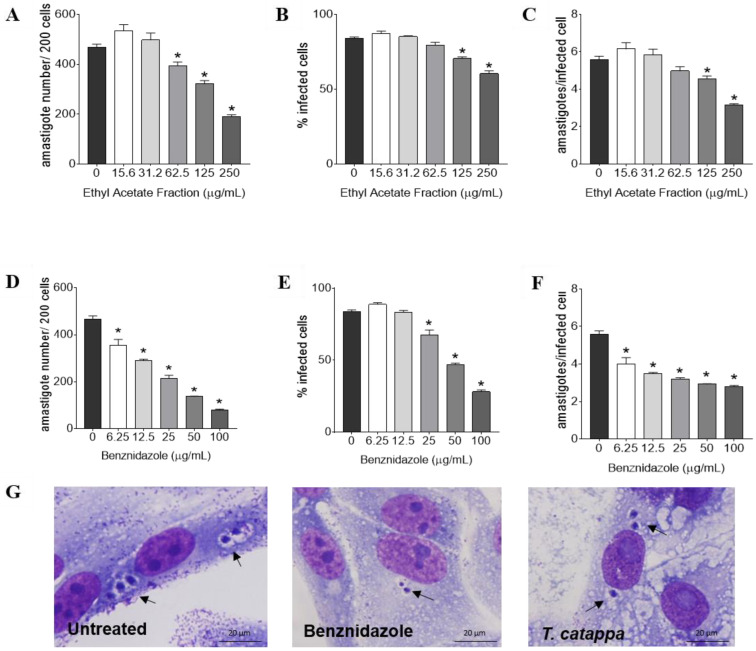
Vero cells infected with *Trypanosoma cruzi* and treated for 24 h with ethyl acetate fraction of *Terminalia catappa* or benznidazole. Parameters of infection (**A**–**F**) and light microscopy (**G**) of cells untreated, treated with benznidazole (100 μg/mL) or treated with ethyl acetate fraction (250 μg/mL). Intracellular amastigotes inside cells (black arrows). The images and data (mean ± SD) represent two independent experiments performed in quadruplicate. * *p* < 0.01 when compared with untreated infected cells by the Mann-Whitney test. Giemsa, objective 100×.

**Figure 4 biology-12-00895-f004:**
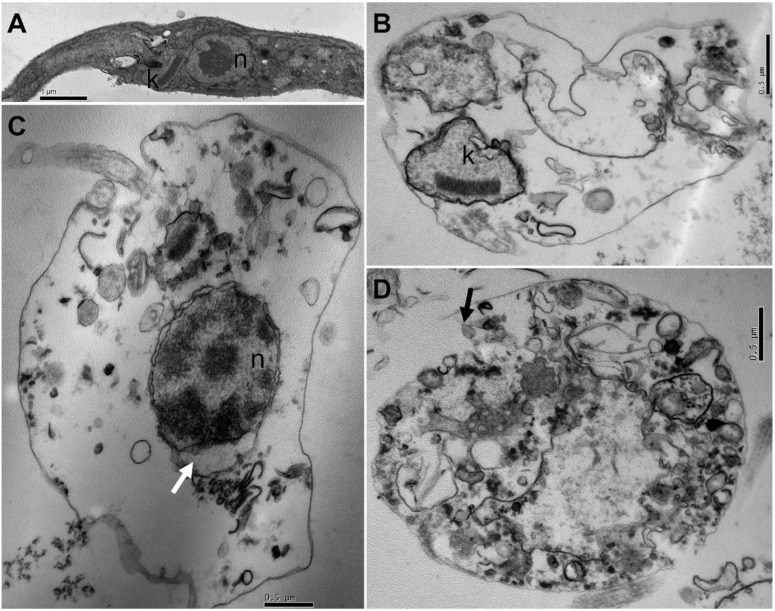
Ultrastructural alterations of *Trypanosoma cruzi* epimastigote forms (**A**) Untreated parasites, with normal characteristics of the protozoan. (**B**–**D**) Parasites treated with *Terminalia catappa* ethyl acetate fraction at 8.8 μg/mL for 72 h. (**C**) Detachment of nuclear membrane (white arrow). (**D**) Severe damage of the cytoplasm and rupture of the plasma membrane (black arrow). n: nucleus; k: kinetoplast.

**Table 1 biology-12-00895-t001:** Compounds identified in *Terminalia catappa*.

	Compound	Mode	Fragments	
1	Hexahydroxydiphenyldigalloylglucose acid	−	MS^2^	785 [M-H]^−^/765; 633 [M-H-152]^−^; 483 [M-H-152-132-18]^−^; 419; 301
			MS^3^	785→483/331 [M-H-152-132-18-152]^−^; 313 [M-H-152-132-18-152-18]^−^; 193; 169
		+	MS^2^	805 [M + H_2_O]^+^/787 [M + H-18]^+^; 617 [M + H-18-152-18]^+^; 467; 303
			MS^3^	805→787/769; 617; 467; 449; 303 [M + H-152-152-162-18]^+^; 277
2	Trigaloyl-hexoside	−	MS^2^	635 [M-H]^−^/483 [M-H-152]^−^; 465 [M-H-152-18]^−^; 313 [M-H-152-18-152]^−^;
			MS^3^	635→465/447; 313 [M-H-152-18-152]^−^; 295 [M-H-152-132-18-18]^−^; 169
		+	MS^2^	659 [M + Na]^+^/489 [M + Na-152-18]^+^; 319
			MS^3^	659→489/471; 337; 319 [M + Na-152-18-152-18]^+^
3	Galoyl-HHDP-hexoside	−	MS^2^	633 [M-H]^−^/463 [M-H-152-18]^−^; 301 [M-H-152-18-162]^−^; 275
			MS^3^	633→301/301 [M-H-152-18-162]^−^
4	Luteolin 8-*C*-hexoside	−	MS^2^	447 [M-H]^−^/357 [M-H-90]^−^; 327 [M-H-120]^−^
			MS^3^	447→327/327 [M-H-120]^−^; 299 [M-H-120-28]^−^
5	Luteolin 6-*C*-hexoside	−	MS^2^	447 [M-H]^−^/429 [M-H-18]^−^; 357 [M-H-90]^−^; 327 [M-H-120]^−^
			MS^3^	447→327/327 [M-H-120]^−^; 299
6	Apigenin 8-*C*-hexoside	−	MS^2^	431 [M-H]^−^/341 [M-H-90]^−^; 311 [M-H-120]^−^; 283 [M-H-120-18]^−^
			MS^3^	431→311/311 [M-H-120-18]^−^; 283
7	Luteolin-6-*C*-(2″-galloyl)-hexoside	−	MS^2^	599 [M-H]^−^/555 [M-H-44]^−^; 447 [M-H-152]^−^; 429 [M-H-152-18]^−^; 327 [M-H-152-120]^−^
			MS^3^	599→447/429; 357; 327 [M-H-152-120]^−^
		+	MS^2^	601 [M + H]^+^/583 [M + H-18]^+^; 481 [M + H-120]^+^; 449; 431 [M + H-170]^+^; 383; 329; 311
			MS^3^	601→431/413;395; 383; 353; 311 [M + H-18-152-120]^+^
8	Apigenin 6-*C*-hexoside	−	MS^2^	431 [M-H]^−^/413 [M-H-18]^−^; 341 [M-H-90]^−^; 311 [M-H-120]^−^
			MS^3^	431→311/311; 283 [M-H-120-28]^−^
9	Apigenin 6-*C*-(2″-galloyl)-hexoside	−	MS^2^	583 [M-H]^−^/431 [M-H-152]^−^; 413 [M-H-152-18]^−^; 311 [M-H-152-120]^−^; 169
			MS^3^	583→431/341; 311 [M-H-152-120]^−^; 283
		+	MS^2^	585 [M + H]^+^/567; 465; 415 [M + H-170]^+^; 379; 367; 313
			MS^3^	585→415/397; 379 [M + H-170-18-18]^+^; 325; 295; 283
10	Quercetin 3-*O*-hexoside	−	MS^2^	463 [M-H]^−^/343 [M-H-120]^−^; 301 [M-H-162]^−^
			MS^3^	463→301/301 [M-H-162]^−^; 179; 151
		+	MS^2^	465 [M + H]^+^/303 [M + H-162]^+^
11	Ellagic acid	−	MS^2^	301 [M-H]^−^/301
		+	MS^2^	627 [2M + Na]^+^/325 [2M + Na-302]^+^
12	Kaempferol 3-*O*-(6″-deoxyhexosyl) -hexoside	−	MS^2^	593 [M-H]^−^/447 [M-H-146]^−^; 429 [M-H-146-18]^−^; 309 [M-H-146-18-120]^−^; 285 [M-H-146-162]^−^
			MS^3^	593→429/309 [M-H-146-18-120]^−^
		+	MS^2^	595 [M + H]^+^/577; 449; 287 [M + H-146-162]^+^

**Table 2 biology-12-00895-t002:** Antioxidant activity of the *Terminalia catappa* extract and fractions.

*Terminalia catappa*	DPPH IC_50_ (µg/mL)	FRAP (µM TE/mg Extract)
Hydroalcoholic extract	221.70 ± 2.43	953.02 ± 0.16
Hexanic fraction	84.77 ± 1.05	153.52 ± 0.02
Ethyl acetate fraction	7.77 ± 1.61	687.61 ± 0.26
Aqueous fraction	5.26 ± 1.26	1009.32 ± 0.13
Trolox	12.80 ± 1.09	-

DPPH: 2,2-Diphenyl-1-picrylhydrazyl radical scavenging assay; IC_50_: inhibitory concentration of 50%; FRAP: ferric reducing antioxidant power assay; TE: Trolox equivalent.

**Table 3 biology-12-00895-t003:** Cytotoxicity, antitrypanocidal activity and selectivity index of *Terminalia catappa*.

*T. catappa*	Cytotoxicity CC_50_ (µg/mL)	*T. cruzi* IC_50_ (µg/mL)					
	Vero	Epimastigote	SI_epi_	Tripomastigote	SI_tri_	Intracellular Amastigote	SI_ama_
Hydroalcoholic extract	>1000	70.85 ± 1.10	>14.11	25.42 ± 1.37	>39.33	-	-
Hexanic fraction	222.40 ± 1.51	148.40 ± 1.20	1.49	34.51 ± 1.15	6.44	-	-
Ethyl acetate fraction	>1000	8.86 ± 1.13	>112.86	24.91 ± 1.15	>40.14	85.01 ± 1.21	>11.76
Aqueous fraction	>1000	104.50 ± 1.11	>9.56	54.59 ± 1.13	>18.31	-	-
Ellagic acid	>1000	215.18 ± 1.57	>4.64	>500	nd	-	-
Benznidazole	>1000	8.66 ± 1.22	>115.47	9.29 ± 1.28	107.64	12.84 ± 1.08	77.88

CC_50_: inhibitory concentration of 50% of cells; IC_50_: inhibitory concentration of 50% of parasites; SI_epi_: selective index of cells (CC_50_) over epimastigote IC_50_; SI_ama_: selective index of cells (CC_50_) over intracellular amastigote IC_50_.

## Data Availability

Data are contained within the article.

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
