# Peer review of "In Vitro Antioxidant and Antitrypanosomal Activities of Extract and Fractions of Terminalia catappa"

_biology, 2023, doi:10.3390/biology12070895_

Round 1
Reviewer 1 Report
The article written by Sandra Alves de Araújo et al. presents in vitro antioxidant and antitrypanosomal activities of extract and fractions of Terminalia catappa. The article is very well written and interesting. The manuscript can be accepted for publication after major revision. The authors should revise the manuscript according to the following comments.
1. What is the main ingredient of hydroalcoholic extract.
2. As you wrote Ellagic acid was identified as the major compound in the ethyl acetate fraction. However, the ethyl acetate fraction is more active against Epimastigote and Tripomastigote than pure Ellagic acid. Which other ingredient causes this fraction to be this active.
Author Response
We thank you the reviewer for the time spent in our mansucript. Find below the point-by-point answer to each comment.
The article written by Sandra Alves de Araújo et al. presents in vitro antioxidant and antitrypanosomal activities of extract and fractions of Terminalia catappa. The article is very well written and interesting. The manuscript can be accepted for publication after major revision. The authors should revise the manuscript according to the following comments.
- What is the main ingredient of hydroalcoholic extract.
ANSWER: We only can perform a relative quantification of the detected compounds by the mass spectrometry, which identified as compounds in highest relative concentration the punicalin and punicalagin, which are compounds derived from ellagic acid, and also luteolin, apigenin and quercetin.
- As you wrote Ellagic acid was identified as the major compound in the ethyl acetate fraction. However, the ethyl acetate fraction is more active against Epimastigote and Tripomastigote than pure Ellagic acid. Which other ingredient causes this fraction to be this active.
ANSWER: Synergistic effects between ellagic acid and other compounds may have been responsible for the antitrypanosomal action of the ethyl acetate fraction. For example, luteolin, apigenin or quercetin, which were also identified in the fraction. This possible synergistic effect was discussed in the manuscript (line 315-316).
Reviewer 2 Report
This study provides information on the in vitro antioxidant and antitrypanosomal activities of extracts and fractions of a medicinal plant, Terminalia catappa. The extract and fractions in vitro -tested showed activity against the three forms, epimastigotes, trypomastigotes and intracellular amastigotes of T. cruzi, the causative agent of Chagas disease. From the results, the authors conclude that the ethyl acetate fraction obtained from Te. catappa leaves can be an effective alternative in the treatment and control of Chagas disease. There are few reports of plant activity against T. cruzi intracellular amastigote forms; therefore, it is of interest to test the efficacy of Te. catappa against the parasite. This study provides evidence that Te. catappa leaves could be an effective alternative in the treatment and control of Chagas disease. The study design is fine and the results are convincing. The following are specific minor remarks to be considered:
- To avoid confusion of the genus names, Trypanosoma and Terminalia, replace T. catappa, by Te. catappa throughout the text, including Table 3, Fig. 2G.
- Line 286-287; replace T. avicenioides, T. leiocarpa and T. molis by Te. avicenioides, Te. leiocarpa and Te. molis
- Line 323; replace T. ivorensis by Te. ivorensis
- Line 326; replace T. albida, by Te. albida
- Line 467; replace leishmania amazonensis by Leishmania amazonensis
- Lines 527-529; incomplete citation, check thoroughly and correct. 41. Ohashi, M.; Amoa-Bosompem, M.; Kwofie, K. D.; Agyapong, J.; Adegle, R.; Sakyiamah, M. M.; ... & Ohta, N. In vitro antiprotozoan activity and mechanisms of action of selected G hanaian medicinal plants against trypanosoma, leishmania, and plasmodium parasites. Phytother. res. 2018, 32(8), 1617-1630. https://doi.org/10.1002/ptr.6093.
- To avoid confusion of the genus names, Trypanosoma and Terminalia, replace T. catappa, by Te. catappa throughout the text, including Table 3, Fig. 2G.
- Line 286-287; replace T. avicenioides, T. leiocarpa and T. molis by Te. avicenioides, Te. leiocarpa and Te. molis
- Line 323; replace T. ivorensis by Te. ivorensis
- Line 326; replace T. albida, by Te. albida
- Line 467; replace leishmania amazonensis by Leishmania amazonensis
- Lines 527-529; incomplete citation, check thoroughly and correct. 41. Ohashi, M.; Amoa-Bosompem, M.; Kwofie, K. D.; Agyapong, J.; Adegle, R.; Sakyiamah, M. M.; ... & Ohta, N. In vitro antiprotozoan activity and mechanisms of action of selected G hanaian medicinal plants against trypanosoma, leishmania, and plasmodium parasites. Phytother. res. 2018, 32(8), 1617-1630. https://doi.org/10.1002/ptr.6093.
Author Response
We thank you the reviewer for the time spent in our mansucript. Find below the point-by-point answer to each comment.
- To avoid confusion of the genus names, Trypanosoma and Terminalia, replace T. catappa, by Te. catappa throughout the text, including Table 3, Fig. 2G.
ANSWER: We replaced T. catappa, by Te. Catappa in the manuscript as requested.
- Line 286-287; replace T. avicenioides, T. leiocarpa and T. molis by Te. avicenioides, Te. leiocarpa and Te. Molis
ANSWER: Altered in the manuscript.
- Line 323; replace T. ivorensis by Te. Ivorensis
ANSWER: Altered in the manuscript.
- Line 326; replace T. albida, by Te. Albida
ANSWER: Altered in the manuscript.
- Line 467; replace leishmania amazonensis by Leishmania amazonensis
ANSWER: Altered in the manuscript.
- Lines 527-529; incomplete citation, check thoroughly and correct. 41. Ohashi, M.; Amoa-Bosompem, M.; Kwofie, K. D.; Agyapong, J.; Adegle, R.; Sakyiamah, M. M.; ... & Ohta, N. In vitro antiprotozoan activity and mechanisms of action of selected G hanaian medicinal plants against trypanosoma, leishmania, and plasmodium parasites. Phytother. res. 2018, 32(8), 1617-1630. https://doi.org/10.1002/ptr.6093.
ANSWER: Complete citation was added in the manuscript.
Reviewer 3 Report
In this article, a study of the antitrypanosomicidal and antioxidant activity of extracts and fractions of Terminalia catappa was carried out.
For this reviewer, studies that focus on neglected diseases such as Chagas disease are of great importance, since the greatest effort to find new treatment alternatives comes mainly from the academy.
In general, the study is well conducted and with interesting results, since from the ethyl acetate fraction there were activities comparable to beznidazole in -epi, -tripo and amastigotes.
I only have a few small appreciations:
In section 3.1, a fractionation was made with hexane, ethyl acetate and a water/methanol mixture, but the chemical characterization was done only with the hydroalcoholic and ethyl acetate extract. Why are the results of the hexanic extract not shown?
In the trypanosomicidal activity and cytotoxicity assays, the final percentage of solvent per well is not mentioned.
Undoubtedly, the article deserves to be published in this journal only considering these two appreciations.
Kind regards
Author Response
We thank you the reviewer for the time spent in our mansucript. Find below the point-by-point answer to each comment.
In this article, a study of the antitrypanosomicidal and antioxidant activity of extracts and fractions of Terminalia catappa was carried out.
For this reviewer, studies that focus on neglected diseases such as Chagas disease are of great importance, since the greatest effort to find new treatment alternatives comes mainly from the academy.
In general, the study is well conducted and with interesting results, since from the ethyl acetate fraction there were activities comparable to beznidazole in -epi, -tripo and amastigotes.
I only have a few small appreciations:
In section 3.1, a fractionation was made with hexane, ethyl acetate and a water/methanol mixture, but the chemical characterization was done only with the hydroalcoholic and ethyl acetate extract. Why are the results of the hexanic extract not shown?
ANSWER: In the chemical evaluation, we focused on the fraction that exhibited the best activity, the ethyl acetate fraction. We displayed the chromatograms of the hydroalcoholic extract , that originated all the fractions, and of the ethyl acetate fraction to demonstrate the difference in components relative quantification, which may be associated to the activity showed by the ethyl acetate fraction. We believe that it was a more objective way to compare the extract and the active fraction.
In the trypanosomicidal activity and cytotoxicity assays, the final percentage of solvent per well is not mentioned.
ANSWER: Thank you for your comment. The solvent DMSO as used at final concentration lower than 1%. This information was added in the manuscript (line 99-101).
Undoubtedly, the article deserves to be published in this journal only considering these two appreciations.
ANSWER: We thank you for your careful review.
Round 2
Reviewer 1 Report
I accept the manuscript for publication in its present form.